

# Bacterial microbiota composition of *Ixodes ricinus* ticks: the role of environmental variation, tick characteristics and microbial interactions

Tuomas Aivelo[1,2], Anna Norberg[1] and Barbara Tschirren[3]

[1] Organismal and Evolutionary Biology research program, University of Helsinki, Helsinki, Finland
[2] Department of Evolutionary Biology and Environmental Studies, University of Zurich, Zurich, Switzerland
[3] Centre for Ecology and Conservation, University of Exeter, Penryn, United Kingdom

## ABSTRACT

Ecological factors, host characteristics and/or interactions among microbes may all shape the occurrence of microbes and the structure of microbial communities within organisms. In the past, disentangling these factors and determining their relative importance in shaping within-host microbiota communities has been hampered by analytical limitations to account for (dis)similar environmental preferences ('environmental filtering'). Here we used a joint species distribution modelling (JSDM) approach to characterize the bacterial microbiota of one of the most important disease vectors in Europe, the sheep tick *Ixodes ricinus*, along ecological gradients in the Swiss Alps. Although our study captured extensive environmental variation along elevational clines, the explanatory power of such large-scale ecological factors was comparably weak, suggesting that tick-specific traits and behaviours, microhabitat and -climate experienced by ticks, and interactions among microbes play an important role in shaping tick microbial communities. Indeed, when accounting for shared environmental preferences, evidence for significant patterns of positive or negative co-occurrence among microbes was found, which is indicative of competition or facilitation processes. Signals of facilitation were observed primarily among human pathogens, leading to co-infection within ticks, whereas signals of competition were observed between the tick endosymbiont *Spiroplasma* and human pathogens. These findings highlight the important role of small-scale ecological variation and microbe-microbe interactions in shaping tick microbial communities and the dynamics of tick-borne disease.

# INTRODUCTION

Microbial communities within organisms consist of symbionts, commensals, mutualists and pathogens that co-occur simultaneously and potentially influence each other (*Petney & Andrews, 1998*; *Rigaud, Perrot-Minnot & Brown, 2010*; *Sofonea, Alizon & Michalakis, 2015*). These microbial communities may be shaped by a range of factors and processes,

Corresponding author
Tuomas Aivelo,
tuomas.aivelo@helsinki.fi

including the environment, host and microbe genetics and the occurrence and abundance of other microbial species (*Adair & Douglas, 2017*). For example, certain microbial species might tolerate only specific abiotic conditions, which makes it more likely that species with similar requirements co-occur within a host ('environmental filtering', *Dallas & Presley, 2014*). Similarly, the host's immune system can influence colonization success of microbes (*Hawley & Altizer, 2011*), with cross-immunity preventing the colonization of different microbes with similar antigenic properties (*Durand et al., 2015*). Furthermore, mutualistic interactions between hosts and microbes can influence the structure of bacterial communities within host individuals (*Chu & Mazmanian, 2013*; *Lee et al., 2013*). Finally, direct interactions among microbes might affect colonization, or replication success after colonization, through competition or facilitation processes. Competition may occur when different microbes use the same, limited resources within a host (*Lello et al., 2004*), whereas facilitation may occur directly through the production of public goods (*West & Buckling, 2003*) or indirectly through the modification of the host's physiology (*Abraham et al., 2017*) or immune defense (*Rodríguez et al., 1999*).

*Ixodes ricinus* is the most common tick species in Europe and an important vector for a range of human, domestic animal and wildlife pathogens (*Medlock et al., 2013*). Its distribution and abundance are strongly influenced by environmental conditions, in particular temperature and humidity (*Cortinas et al., 2002*; *Gatewood et al., 2009*). Previous studies that characterized the bacterial community composition of *I. ricinus* ticks have found mostly environmental and free-living bacteria but also several endosymbionts and human, domestic animal or wildlife pathogens, including *Borrelia* (*Mannelli et al., 2012*), *Rickettsia* (*Venclikova et al., 2014*), *Anaplasma* (*Jahfari et al., 2014*) and *Candidatus* Neoehrlichia (*Kawahara et al., 2004*).

Differences in the bacterial community structure and composition of ticks across habitats (*Estrada-Peña et al., 2018*), geographical sites (*Carpi et al., 2011*), and tick life stages and sexes (*Carpi et al., 2011*; *Vayssier-Taussat et al., 2013*) have been documented. Large-scale biotic or abiotic factors such as vegetation structure, elevation, temperature or rainfall may influence tick microbial communities directly, or indirectly through effects on tick physiology or activity patterns (*Van Treuren et al., 2015*) or via influencing the distribution and abundance of tick hosts species (*Randolph et al., 1999*; *MacDonald et al., 2017*). Small-scale and/or tick-specific effects on microbial communities may be explained by microhabitat or microclimatic conditions experienced by individual ticks (*Gern, Morán Cadenas & Burri, 2008*), individual tick behavior or genetics (*Hawlena et al., 2013*), direct biotic interactions among microbes (*Moutailler et al., 2016*) or parallel acquisition from a host during a bloodmeal (*Andersson, Scherman & Råberg, 2014*; *Belli et al., 2017*; *Swei & Kwan, 2017*).

Currently, the relative importance of these factors in shaping tick microbial communities is not well understood, which hampers progress in our understanding of the processes shaping microbial communities in nature and predicting the occurrence of specific microbes (e.g., human pathogens). Elevational gradients are excellently suited to quantify the importance of large-scale ecological variation in shaping tick bacterial microbiota because they cover a large range of environmental conditions within a small geographical

area. Furthermore, including replicated transects along gradients allow us to quantify the robustness of ecological associations within sites and along elevational clines on tick microbial communities.

*Ixodes* ticks are commonly found to be co-infected with several (human, domestic animal and/or wildlife) pathogens (*Andersson, Scherman & Råberg, 2013*; *Michelet et al., 2014*; *Diuk-Wasser, Vannier & Krause, 2016*; *Moutailler et al., 2016*). Currently, it is unknown whether these co-infection patterns are caused by similar environmental preferences of pathogens, parallel acquisition from host communities or direct microbe-microbe interactions within ticks. Yet, previous studies suggest that the latter process, (i.e., facilitation and competition processes among microbes) may play a role in shaping microbial communities (*Haine, 2008*; *Bonnet et al., 2017*). For example, it has been found that pathogenic *Rickettsia* species prevent co-infection with other *Rickettsia* species in *Dermacentor variabilis* ticks (*Macaluso et al., 2002*), whereas the presence of *Francisella* sp. endosymbionts increases the colonization success of pathogenic *Francisella novicida* in *D. andersoni* ticks (*Gall et al., 2016*). Facilitation has also been suggested to promote co-infection with different *Borrelia afzelii* strains in *Ixodes ricinus* ticks (*Andersson, Scherman & Råberg, 2013*). Most strikingly, dysbiosis in *I. scapularis* ticks (i.e., ticks with low microbiotal diversity) leads to a defective peritrophic matrix which decreases the colonization success of *B. burgdorferi* s.s., suggesting that the pathogen requires the presence of an intact microbiota to be able to invade ticks (*Narasimhan et al., 2014*). Thus, the microbial community may have a crucial impact on vector competence of ticks and thereby on disease dynamics.

Yet, as outlined above, co-occurrence of microbes can be due to environmental filtering or direct microbial interactions, and distinguishing between these processes is non-trivial. Indeed, previous studies that have documented pathogen co-occurrence in ticks have not accounted for potential confounding variables such as shared ecological requirements, and are thus limited in their ability to differentiate between co-occurrences due to shared environmental niches, and co-occurrence shaped by facilitation or competition among microbes.

To address these gaps, we exploited the substantial environmental heterogeneity along replicated elevational gradients in the Swiss Alps to quantify the relative importance of environmental factors, tick characteristics and direct microbial interactions in influencing the structure of bacterial communities in *I. ricinus* ticks in general, and the (co-)occurrence of pathogens in particular, using a combination of 16S sequencing and joint species distribution modelling (JSDM) (*Warton et al., 2015*). By taking shared environmental preferences into account, JSDMs allows to identify residual co-occurrence patterns among microbes that can result from unaccounted environmental effects or direct microbial interactions. However, the correct spatial scale with regards to the focal biological processes is of importance, as well as the type of the hypothesized biotic interaction (*Araújo & Rozenfeld, 2014*; *Zurell, Pollock & Thuiller, 2018*) when interpreting JSDM patterns (*Dormann et al., 2018*).

Specifically, we ask (i) how do large-scale abiotic factors and small scale tick-level variables affect tick microbiota composition, (ii) which large-scale abiotic and small-scale tick-level variables predict pathogen occurrence, and (iii) are there patterns of non-random

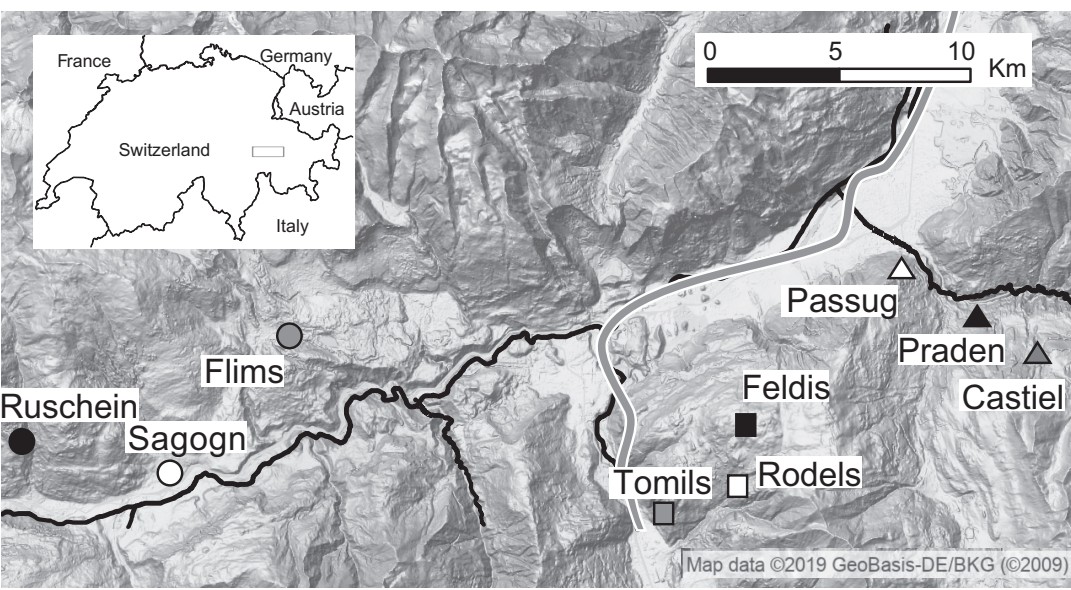

**Figure 1** **Location of tick sampling sites in the Swiss Alps.** Different shapes (i.e., circle, square and triangle) represent the different locations and different colours represent elevation (white: low, grey: middle, black: high). Rivers and motorway are shown in black. Map data ©2019 Google, GeoBasis-DE/BKG.

microbial co-occurrence that cannot be explained by environmental responses and might be due to unmeasured variables, such as microbial interactions.

# MATERIAL AND METHODS

## Tick sampling

Questing *Ixodes ricinus* ticks were collected at three locations in the Swiss Alps (Kanton Graubünden). At each location, one site at low (630–732 m above sea level, masl), one at medium (1,094–1,138 masl) and one at high (1,454–1 673 masl) elevation were identified (Fig. 1, Table 1, *N* = 9 sampling sites). At each site, questing ticks were sampled thrice, once in June, once in July, and once in August 2014 by dragging a white blanket (1 m × 1 m) over the ground vegetation as described previously (*Lemoine, Cornetti & Tschirren, 2018*). Ticks were collected from the blanket and stored in 95% ethanol. Tick species, sex and life stage were verified by morphological features following (*Hillyard, 1996*) using a stereomicroscope.

## Environmental variables

For each sampling site, we compiled information on large-scale, site-level ecological variables by obtaining data on elevation, slope and aspect using DHM25, land use data from swissTLM3D (both from Federal Office of Topography swisstopo) and data on temperature and precipitation from Landscape Dynamics (Swiss Federal Research Institute for Water, Snow and Landscape Research WSL and Federal Office of Meteorology and Climatology MeteoSwiss; *Thornton, Running & White, 1997*). Data on *I. ricinus* abundance and the abundance of a key tick host, the bank vole (*Myodes glareolus*), as well as the

**Table 1  Tick sampling sites in the Swiss Alps.**

| Location | Site | Coordinates | | Elevation | Succesfully sequenced *Ixodes ricinus* ticks | | |
|---|---|---|---|---|---|---|---|
| | | North | East | (masl) | nymphs | males | females |
| 1 | Sagogn | 46.783 | 9.233 | 693 | 0 | 9 | 15 |
| | Flims | 46.827 | 9.280 | 1138 | 3 | 5 | 3 |
| | Ruschein | 46.795 | 9.169 | 1454 | 0 | 1 | 1 |
| 2 | Rodels | 46.760 | 9.425 | 630 | 2 | 5 | 4 |
| | Tomils | 46.772 | 9.453 | 1144 | 3 | 6 | 4 |
| | Feldis | 46.789 | 9.453 | 1673 | 1 | 1 | 0 |
| 3 | Passug | 46.840 | 9.538 | 732 | 0 | 5 | 6 |
| | Castiel | 46.826 | 9.569 | 1094 | 0 | 3 | 3 |
| | Praden | 46.817 | 9.589 | 1582 | 1 | 0 | 1 |

ratio of bank vole to other rodents at our sampling sites were obtained from *Cornetti et al. (2016)* and *Cornetti et al. (2018)*. Details on the different variables and a justification why they were included to characterise large-scale ecological conditions is provided in the Supplemental Information.

## Tick DNA isolation and quantification of neutral genetic diversity

The number of analysed ticks per site is presented in Table 1. Although we aimed to include similar numbers of *I. ricinus* ticks from each sampling site and month, variation in the number of ticks per site (Table 1) was unavoidable because of variation in tick abundance across sites (*Lemoine, Cornetti & Tschirren, 2018*). To avoid contamination, we performed DNA isolation and amplifications in a laminar flow cabinet. Each tick was washed thrice with sterile water before sterilizing it with 3% hydrogen peroxide. Ticks were then cut in half with a sterilized blade to facilitate DNA isolation. DNA was extracted using DNeasy Blood & Tissue kit (Qiagen; Hilden, Germany).

Host genetics may affect pathogen and endosymbiont colonisation and replication success (*Archie & Ezenwa, 2011*). In order to quantify individual and population-level genetic diversity, we genotyped ticks at 11 microsatellite markers in two multiplexed amplifications (see Supplementary Material for details). Not all markers were successfully amplified in all samples, but none of the samples contained more than two failed markers. We used package *poppr* (*Kamvar, Tabima & Grünwald, 2014*) in R 3.4.1 (*R Core Team, 2013*) to test for linkage disequilibrium and deviation from Hardy-Weinberg equilibrium. Individual observed heterozygosity was determined for each tick as a proportion of heterozygous markers to all successfully amplified markers. Expected population level heterozygosity was determined with *poppr*. The former was used as a tick-level explanatory variable (together with tick sex and life stage), the latter was used as a site-level explanatory variable.

## Tick microbiota sequencing

We characterized tick bacterial community composition by sequencing the hypervariable V4 region of the 16S rRNA (16S) gene. Negative controls (extraction reagent blank, $N = 2$ and PCR controls, $N = 3$) were processed alongside the tick samples. Sequencing libraries

were prepared following the Earth Microbiome 16S Illumina Amplicon protocol, using the primers 515FB and 806RB (*Carey, Walters & Knight, 2013*) (see Supplemental Information for details). Samples and negative controls were randomized across two plates. The libraries were sequenced on Illumina MiSeq at the Functional Genomic Center Zurich with a target length of 250 bp following the manufacturer's protocol. The obtained sequence data were analyzed following the *mothur* pipeline with MiSeq standard operation procedures (*Kozich et al., 2013*). Sequences have been deposited to the Sequence Read Archive under BioProject PRJNA506875. The complete metadata of the samples and their matching sequence accession numbers have been submitted to FigShare (doi:10.6084/m9.figshare.7380767).

As we are not able to assess whether individual OTUs are resident or not, and we do not know their transmission routes, a special focus of our analysis was on tick endosymbionts and tick-borne human, domestic animal or wildlife pathogens (Table 2), which are obligate residents. This approach does not mean that the other OTUs would not have a substantial effects on ticks and other tick symbionts. Identification of endosymbionts and pathogens is described in the Supplemental Information.

## Joint species distribution modelling of microbiota composition

Only samples with >500 reads and OTUs which were present in at least two samples were included in the analyses (Table 1). As the most common OTU, the intra-mitochondrial endosymbiont *Candidatus* Midichloria (*Lo et al., 2006*), was present in all samples, it was not included in the modelling. For the occurrence matrix, an OTU was determined to be present in a tick if >5 reads were identified in a sample (following *Aivelo & Norberg, 2017*).

We used a JSDM framework called Hierarchical Modelling of Species Communities (HMSC, (*Ovaskainen et al., 2017a*) to examine how environmental variables correlate with pathogen and tick endosymbiont occurrence in ticks, and whether there are non-random residual associations among different OTUs and/or oligotypes, implying direct facilitation or competition effects among microbes. This approach combines information on environmental covariates, bacterial species traits, spatiotemporal context and sampling design to explain the presence or absence of OTUs (Fig. S2). The associations among OTUs are captured with the latent part of the framework, modelling the residual variance after accounting for the effects of the environment. The estimates for these latent variables can be then translated into residual correlations among response variables, i.e., OTUs and/or oligotypes. These correlations thus reflect (dis)associations which cannot be explained by shared responses to the environment.

We compiled occurrence matrices for OTUs for each individual tick as a response variable. For each sampling unit, i.e., a row in our response variable matrix, we included information on the identity of the sampling unit (tick ID), its location, sampling site (for which we included also the spatial structure as coordinates) and month, describing the study design. To reach a better resolution within specific OTUs, we analyzed known human, domestic animal or wildlife pathogens, tick endosymbionts and their close relatives within the 100 most common OTUs with oligotyping pipeline (*Eren et al., 2014*). Oligotyping uses all the sequences, which form an OTU, and performs Shannon Entropy Analysis to regroup sequences based on within-OTU variation. This results in higher-resolution

**Table 2  Common tick endosymbionts and/or putative human pathogens observed in *I. ricinus* ticks.**
See Supplemental Information for information on OTU assignment.

| OTU | Label | Human pathogen/ tick endosymbiont | Occurrence (% of analyzed ticks) |
|---|---|---|---|
| Otu0001 | *Midichloria* | endosymbiont | 100 |
| Otu0003 | *Spiroplasma* | endosymbiont | 41 |
| Otu0005 | *Rickettsiella* | endosymbiont | 63 |
| Otu0021 | *Lariskella* | endosymbiont | 49 |
| Otu0031 | *Rickettsia helvetica* | both | 16 |
| | *R. monacensis* | both | 6 |
| Otu0067 | *Rickettsia sp.* | both | 25 |
| Otu0076 | *Anaplasma* | both | 33 |
| Otu0086 | *Candidatus* Neoehrlichia | both | 22 |
| Otu0088 | *Borrelia afzelii* | pathogen | 9 |
| | *B. miyamotoi* | pathogen | 10 |
| | *B. garinii* | pathogen | 6 |
| | *B. valaisiana* | pathogen | 2 |

grouping than OTUs as the different oligotypes might differ only by a single nucleotide (*Eren et al., 2014*). We used the standard operation procedures of the oligotyping pipeline software (http://oligotyping.org; *Eren et al., 2013*). We labelled the resulting oligotypes through BLAST search (*Camacho et al., 2009*). For some species, such as *Rickettsia* spp., the V4 region of 16S might not have enough resolution (*Greay et al., 2018*), and thus, the labels should not be considered as definite identifications.

Including a large number of explanatory variables in statistical models is inherently challenging. To reduce the number of variables, while maintaining their information value, we used two variable sets in the model: (a) a set of full-effect explanatory variables, and (b) explanatory variables under variable selection (*Ovaskainen et al., 2017b*). The full-effect variable set included an intercept, two tick-level variables (tick sex or life stage and individual heterozygosity) and two site-level variables (tick abundance and elevation of the sampling site). Additionally, we included information whether a specific OTU is an endosymbiont and/or a human, domestic animal or wildlife pathogen (*Abrego, Norberg & Ovaskainen, 2016*). This allowed us to test if endosymbionts and/or pathogens respond differentially to environmental conditions than other OTUs. The set of explanatory variables under variable selection included additional information on the environmental conditions of the sites (namely the number of days above 7 °C during the year, monthly precipitation, mean monthly temperature, forest coverage, slope, aspect, bank vole abundance, the proportion of voles to other rodents and expected tick heterozygosity) (Table S1). We considered all parameter estimates, including associations among bacterial OTUs, having strong statistical support and thus being 'significant' if the 90% central credible interval of the parameter did not overlap with zero (see Supplemental Information for additional model details). The model was run in Matlab R2017 (The MathWorks, Natick, MA, USA).

## RESULTS

### Ixodes ricinus *microbiota composition*

We 16S sequenced the bacterial community of 92 *Ixodes ricinus* ticks which resulted in 13,214,477 reads. No amplification was observed in the five negative controls (i.e., their sequencing did not result in any reads) and one tick was sequenced twice. After contig assembly and quality control 1,656,287 reads were retained. Most of the discarded reads were either shorter than 250 bp or chimeras. There was a median of 1,562 quality-controlled reads per sample, with an interquartile range of 6319. 82 samples with more than 500 reads per sample, a plateauing accumulation curve and a Good's coverage estimator ≥0.95 were included in the subsequent analyses (Fig. S1). In total, 5,181 bacterial OTUs were identified. The median number of OTUs when rarified to 500 reads per sample was 89 OTUs, with a 95% confidence interval of 78.3–98.5 OTUs.

Six OTUs were present in at least 90% of the samples: *Ca.* Midichloria (Otu0001), *Sphingomonas* (Otu0002, 0006 and 0007), *Pseudomonas* (Otu0011) and *Delftia* (Otu0012). Together, they represented 50.2% of all reads. We used oligotyping to further divide OTU0031 '*Rickettsia*' into two oligotypes labelled as '*R. helvetica*' and '*R. monacensis*', and OTU0086 '*Borrelia*' into four oligotypes labelled as '*B. afzelii*', '*B. valaisian* a' and '*B. garinii*' and '*B. miyamotoi*'. 635 OTUs and oligotypes were used in subsequent analyses, including 14 endosymbionts and / or human, domestic animal or wildlife pathogens (Table 2).

### Tick microbiota variance partitioning

Variance partitioning revealed that most of the variation in tick microbiota composition explained by our model related to tick ID: for the hundred most common OTUs, tick ID accounted for 64.1% of the variation explained by the model. Fixed effects (e.g., tick life stage, elevation; see Table S1) accounted for 12.5% (tick-level: 7.3%, site-level: 5.2%) and spatial and temporal random effects (i.e., location, site and month) explained 23.3% (Fig. 2). This suggests that there is extensive tick-level variation which cannot be accounted for by tick-specific characteristics included in our model (i.e., sex, life stage, genetic diversity) or site-level environmental factors. The pattern differed slightly for endosymbionts and human, domestic animal or wildlife pathogens: while tick ID was still the most important variable explaining 49.9%, fixed effects explained 31.8% (tick-level: 17.5%, site-level: 14.3%) and spatial and temporal random effects explained 18.3% of the total variation explained by the model, when averaged over all pathogens and endosymbionts (Fig. 2). Thus, tick- and site-level fixed effects explained a larger proportion of the variation in the occurrence of obligate resident pathogens and endosymbionts than the occurrence of other (potentially non-resident) OTUs.

### Tick-specific and environmental factors related to OTU occurrence

The occurrence of tick endosymbionts and pathogens was strongly associated with specific explanatory variables, yet associations were typically microbe-specific rather than universal (Table 3). The two most important variables explaining the presence or absence of tick endosymbionts and human, domestic animal or wildlife pathogens were tick sex and elevation of the sampling site: adult female ticks were less likely to harbour the
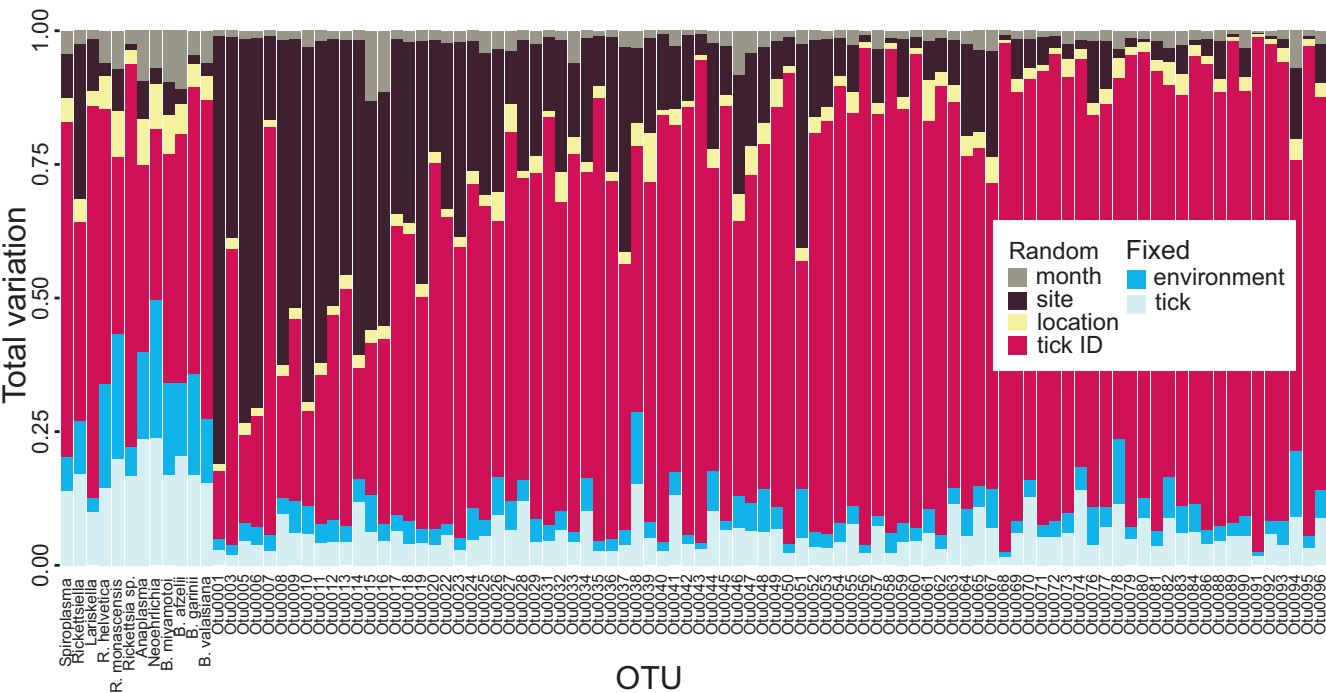

**Figure 2** **Tick microbial community variance partitioning for different fixed and random effects.** The first three columns represent tick endosymbionts, the next three columns are OTUs which are both tick endosymbionts and human pathogens and the subsequent six columns represent human pathogens. The other columns represent the 88 most common OTUs found in I. ricinus, ordered by read frequency. Month, sampling site, location and tick ID were included in the model as random effects, whereas fixed effects were divided into environmental (elevation, temperature, precipitation, forest coverage, slope, aspect, vole abundance and vole-to-other-rodents ratio) and tick-specific variables (life stage or sex, individual heterozygosity, abundance, expected population heterozygosity). See raw data in Figshare for information on OTU labels (DOI: 10.6084/m9.figshare.7380767.v3).

endosymbionts *Spiroplasma*, *Rickettsiella*, *Lariskella* and *Rickettsia* spp. (Table 3), and ticks at higher elevations had higher probability to harbour *R. helvetica* and *R. monacensis*, but were less probable to harbour *B. garinii* (Table 3). Slope and aspect were also significant predictors of tick endosymbionts and pathogen occurrence, with ticks from sites facing northwards having a higher probability of harbouring *Spiroplasma* and *B. afzelii*, and ticks from sites on steeper slopes having a higher probability of harbouring *Rickettsia sp.* (Table 3). Furthermore, a higher tick abundance was associated with a higher probability of *Rickettsiella* and *Ca.* Neoehrlichia occurrence (Table 3). Relationships between tick life stage, mean temperature, the number of days >7 °C or forest cover and the occurrence of specific OTUs were not strongly statistically supported.

The effect sizes of strongly statistically supported associations varied substantially (Figs. S4A–S4I). For example, threefold increase in vole abundance corresponded to less than one percentage point decrease of *R. monacensis* prevalence (Fig. S4B), whereas a threefold increase in tick abundance corresponded to a threefold increase in *Neoehrlichia* prevalence from 8% to 27% (Fig. S4E).

Aivelo et al. (2019), *PeerJ*, DOI 10.7717/peerj.8217

**Table 3  Associations between tick-specific and environmental variables and the occurrence of endosymbionts and human pathogens in *I. ricinus* ticks.** A positive sign indicates that higher variable values are associated with a higher probability of OTU occurrence. A higher aspect value means that a site is facing northwards. Only associations with strong statistical support (based on the 90% central credible interval) are presented.

| | | Full variable set | | | | | Variable selection set | | | | | | | | |
|---|---|---|---|---|---|---|---|---|---|---|---|---|---|---|---|
| | | Tick sex (Female) | Tick life stage (Nymph) | Tick abundance | Tick heterozygosity | Elevation | Tick population expected heterozygosity | Number of days >7 °C | Precipitation | Mean temperature | Forest cover | Slope | Aspect | Vole abundance | Vole/other rodents ratio |
| Otu0003 | *Spiroplasma* | − | | | | | | | | | | | + | | − |
| Otu0005 | *Rickettsiella* | − | | + | − | | − | | + | | | | − | | |
| Otu0022 | *Lariskella* | − | | | − | | | | | | | | | | |
| Otu0031 | *Rickettsia helvetica* | | | | | + | | | | | | | | − | |
| | *R. monacensis* | | | | | + | − | | | | | | | − | |
| Otu0067 | *Rickettsia sp.* | − | | | | | | | | | | | + | | |
| Otu0076 | *Anaplasma* | | | | | | | | | | | | | | |
| Otu0086 | *Ca.* Neoehrlichia | | | + | | | | | | | | | | | |
| Otu0088 | *Borrelia afzelii* | | | | | | | | | | | | + | | |
| | *B. miyamotoi* | | | | | | | | | | | | | | |
| | *B. garinii* | | | | | − | | | | | | | | | |
| | *B. valaisiana* | | | | | | | | | | | | | | |
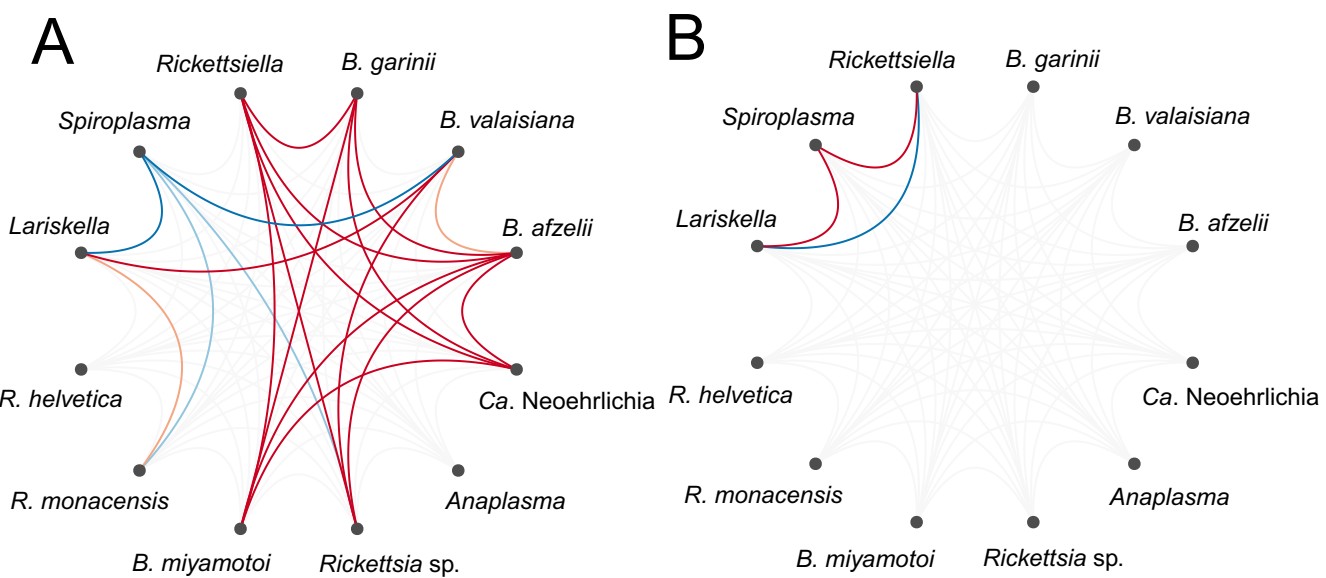

**Figure 3** **Residual association patterns among endosymbionts and human pathogens within ticks on (A) individual tick-level and (B) on site-level after accounting for shared environmental preference.** Red lines represent positive associations and blue lines negative associations. Only associations with strong statistical support (i.e., based on the 90% central credible interval) are presented. Darker colors indicate stronger associations.

### Patterns of microbial association within ticks

Numerous bacterial OTUs were either significantly more or less likely to co-occur within a tick than expected by chance after accounting for shared environmental preferences (Fig. 3A; Table S2). At the level of the individual tick, the occurrence of the tick endosymbiont *Spiroplasma* was negatively associated with the occurrence of the endosymbiont *Lariskella* and several tick-borne pathogens, namely *Rickettsia* sp., *Ca.* Neoehrlichia and *B. miyamotoi* (Fig. 3A). Associations among pathogens, if they occurred, were all positive (Fig. 3A), suggesting that ticks are more likely to be co-infected with several human, domestic animal or wildlife pathogens simultaneously than expected by chance or based on shared environmental preferences. *Borrelia* oligotypes showed positive co-occurrence patterns among each other, except for *B. miyamotoi*, which was not associated with other *Borrelia sp.*, but negatively with *Spiroplasma* and positively with *Lariskella*. At the level of the sampling site, significant associations were sparser. *Spiroplasma* was more likely to co-occur with *Lariskella* and *Rickettsiella* across sites, whereas the latter two were less likely to co-occur across sites than expected by chance after accounting for shared environmental preferences (Fig. 3B). At the level of month or location, there were no significant associations.

### DISCUSSION

We used a JSDM framework to quantify the relative importance of large scale, site-level environmental variables, tick-level characteristics and interactions among microbes in shaping tick microbiota composition along elevational gradients in the Swiss Alps. We show that although our study captured extensive environmental variation, with sampling

sites spanning across an elevational gradient from 630–1,580 masl, and a large number of ecological variables was considered in our models, the explanatory power of such large-scale ecological factors was comparably weak. In contrast, individual tick ID explained over 60% of the variation in microbiota composition. This substantial microbiota variation across individual ticks may be partly explained by some of the bacteria present in ticks being non-resident (i.e., bacteria that were by chance obtained from the environment through the mouth, the anal pore or spiracles or during blood-feeding; *Horner-Devine & Bohannan, 2006*; *Engel & Moran, 2013*; *Zolnik et al., 2016*; *Zolnik et al., 2018*; *Ross et al., 2018*). Indeed, there has been a debate whether ticks have a stable microbiota (*Ross et al., 2018*), mirroring the wider debate on how common resident microbiota is in arthropod hosts (*Hammer et al., 2017*).

However, also for endosymbionts and human, domestic animal or wildlife pathogens, which are obligate resident, tick ID accounted for half of the variation in occurrence, suggesting that microhabitat or -climatic conditions experienced by individual ticks, tick-specific traits and behaviors not included in our models, as well as microbial interactions within ticks such as facilitation and competition (*Abraham et al., 2017*; *Gurfield et al., 2017*), play a crucial role in shaping microbiota composition and the occurrence of endosymbionts and human or wildlife pathogens in *I. ricinus*. Focusing on such small-scale variables, rather than large-scale climatic or environmental factors as is usually done when modelling tick-borne pathogen prevalence (*Norman, Worton & Gilbert, 2016*; *Rosà et al., 2018*), is thus likely a more fruitful approach to advance our understanding of microbiota composition of natural populations as well as (tick-borne) disease dynamics.

Co-occurrence of human, domestic animal or wildlife pathogens in ticks has been documented previously, both in *I. ricinus* (*Lommano et al., 2012*; *Michelet et al., 2014*) and other tick species (*Gurfield et al., 2017*; *Laaksonen et al., 2018*). Yet, previous studies did not control for environmental filtering, which limited their ability to disentangle shared responses to the environment from direct microbe-microbe interactions. Our study revealed that when accounting for shared environmental preferences, associations among human or wildlife pathogens were often pronounced and mostly positive. These positive associations may result from direct facilitation among microbes or parallel colonization from co-infected tick hosts. Because our sampling unit was the whole tick, whereas bacteria inhabiting a tick can be situated in different organs, co-occurrence at the tick-level does not necessarily mean that there is direct interaction between co-occurring OTUs, although indirect interactions, via, e.g., host immune system, can still occur.

Within ticks, the significant positive associations among the Lyme disease-causing *Borrelia* genospecies (*B. afzelii*, *B. garinii* and *B. valaisiana*) were particularly striking. This positive co-occurrence is surprising because *B. garinii* and *B. valaisiana* are bird specialists (*Hanincova et al., 2003b*; *Comstedt, Jakobsson & Bergström, 2011*), whereas *B. afzelii* is a rodent specialist (*Hanincova et al., 2003a*). Thus, the parallel colonization from co-infected tick hosts cannot explain this pattern. Rather the positive co-occurrence is indicative of facilitation processes among *Borrelia* genospecies, as has been suggested previously (*Andersson, Scherman & Råberg, 2013*). Such facilitation, and the resulting co-infection of ticks with several *Borrelia* genospecies has implications for the severity, diagnosis,
treatment and control of Lyme disease. Finally, the co-occurrence of these different *Borrelia* genospecies suggests that *I. ricinus* feeds on multiple, phylogenetically diverse host species during its life cycle and does not show pathogen-mediated host specialization as has been suggested previously (*McCoy et al., 2005*; *McCoy, Léger & Dietrich, 2013*).

Although associations among microbes were mostly positive, there were negative associations between the tick endosymbiont *Spiroplasma* and several human or wildlife pathogens, which may be explained by competition. The most common infection route for *Spiroplasma* is maternal (i.e., vertical) transmission (*Herren & Lemaitre, 2011*), indicating that horizontal or environmental transfer plays a minor role in its transmission. Protective effects of *Spiroplasma* have been previously described in *Drosophila* spp., where *Spiroplasma* is associated with a decreased probability of nematode and parasitoids infections (*Xie, Vilchez & Mateos, 2010*; *Jaenike et al., 2013*). Although the exact mechanisms mediating *Spiroplasma*-induced competition effects are currently unknown, this finding may stimulate further research into the potential of tick endosymbionts to manage tick-borne pathogens.

In contrast to the numerous positive or negative associations among microbes at the tick-level, little statistical support for positive or negative microbial co-occurrence was found at the site-level, with the exception of the associations among three endosymbionts. Interestingly, the pattern of co-occurrence of *Spiroplasma* and *Lariskella* at the site-level was opposite from what was observed at the tick-level. It suggests that *Spiroplasma* and *Lariskella* are more likely to co-occur at the same sites but not within the same tick. Co-occurrence at the site-level can be due to an environmental variable not included in our model, for which the three OTUs had similar responses. It has also been suggested that negative associations generate checkerboard patterns of co-occurrence that can be captured at finer spatial scales but that are lost with increasing scales, but positive associations can be captured across scales (*Araújo & Rozenfeld, 2014*).

Despite the large among-tick variation in microbiota composition, we identified a range of environmental variables that significantly predicted the occurrence of specific tick endosymbionts and human, domestic animal or wildlife pathogens. However, the predictor variables as well as their effect were typically OTU-specific rather than universal. For example, *B. garinii* was *less* likely to occur at higher elevations, whereas *R. helvetica* and *R. monascensis* were *more* likely to occur at higher elevations. Generally, the environmental factors shaping *Rickettsia* spp. distribution are poorly understood, as is their range of host species (*Halos et al., 2010*; *Eremeeva & Dasch, 2015*). Yet, it has previously been found that spotted fever incidence in humans, caused by *R. rickettsii*, is highest in areas or regions, where ticks are less common (*Atkinson et al., 2013*). This is in line with our findings and suggests that *Rickettsia* spp. are more likely to colonize ticks living under suboptimal conditions (e.g., at range edges).

The finding that *B. garinii* is less likely to occur at higher elevations is in line with previous observations (*Jouda, Perret & Gern, 2004b*; *Cornetti et al., 2018*) and may be explained by changes in vegetation structure and associated changes in host communities (*Halos et al., 2010*), in particular changes in the diversity and/or abundance of birds, the natural hosts of *B. garinii* (*Comstedt, Jakobsson & Bergström, 2011*). In contrast, the occurrence of the mammal specialist *B. afzelii* was not related to elevation, potentially because elevational

clines in mammal diversity and/or abundance are less pronounced (*McCain, 2005*). Indeed, we did not observe an association between elevation and bank vole abundance across our study sites (ANOVA: $F_{1,8} = 0.357$, $p = 0.57$, $R^2 = 0.05$).

Interestingly, temperature and precipitation, which vary strongly across elevational gradients (average temperature and precipitation: high elevation sites: 11.8 °C and 17.8 mm per month; in low sites: 16.5 °C and 12.1 mm per month), were not significant predictors of the occurrence of endosymbionts or human or wildlife pathogens, with the exception of precipitation correlating positively with the probability of *Rickettsiella* occurrence. This may be partly explained by the temperature and precipitation measures included in our models not fully capturing the microclimatic variation across sites and along elevational clines. Indeed, slope and aspect, which are important determinants of the topography, and thus microclimate (*Bennie et al., 2008*), were significant predictors of pathogen and endosymbiont occurrence. The probability of *Rickettsia* sp. occurrence was higher on steeper slopes. Furthermore, the probability of occurrence was higher on north-facing slopes for *B. afzelii* and *Spiroplasma* and higher on south-facing slopes for *Rickettsiella* (see also *Stuen, Granquist & Silaghi, 2013*). Microclimatic conditions may affect microbial occurrence directly, or indirectly via affecting tick behavior or host community composition (*Swei, Meentemeyer & Briggs, 2011*; *Lawson et al., 2014*). Furthermore, topography can affect population connectivity and dispersal in metapopulation networks (*Swei & Kwan, 2017*).

Previous work has found that tick abundance is a strong predictor of *Borrelia* spp. prevalence, potentially because larger tick populations facilitate co-feeding transmission (*Jouda, Perret & Gern, 2004a*). No relationship between *Borrelia* spp. occurrence and tick abundance was observed in our study. However, both *Ca.* Neoehrlichia and *Rickettsiella* were more common at sites where ticks were more abundant, suggesting that co-feeding transmission may also play a role in the life cycle of these microbes.

Finally, differences in host competence can lead to dilution effects and thus affect the prevalence of tick-borne pathogens (*Keesing, Holt & Ostfeld, 2006*). Whereas for some tick-borne pathogens the vertebrate hosts are known or suspected (e.g., small mammals for *B. afzelii* (*Hanincova et al., 2003a*) and *Ca.* Neoehrlichia (*Jahfari et al., 2012*), birds for *B. garinii* and *B. valaisiana* (*Hanincova et al., 2003b*), both for *Anaplasma* (*Keesing et al., 2012*) and *R. helvetica* (*Sprong et al., 2009*), for others the host species range is less well understood (e.g., *B. miyamotoi*; *Wagemakers et al., 2015*). The bank vole is a common tick host at our study sites and their abundance was a significant negative predictor of *R. monacensis* and *R. helvetica* occurrence. Interestingly, bank voles are not known hosts for either (*Burri et al., 2014*). Most likely, the relation is thus indirect, explained by an unmeasured biotic or abiotic variable that correlates with bank vole abundance. No evidence was found that the proportion of bank voles to other rodents affects the prevalence of tick-borne pathogens.

A limitation of our sampling design is the uneven sample distribution across sites. We collected ticks up to the upper elevational limit of tick distribution, which leads to a large variation in environmental variables included in our models, but at the same times means that we have a limited number of samples from the high elevation sites. Yet, adequate model fit suggests that this uneven sample distribution did not compromise model performance.

Furthermore, although JSDM is a powerful approach to model community structure, it has a number of limitations. First, it assumes that interactions among microbes are similar across environments (but see *Tikhonov et al., 2017*). This is not necessarily the case as both environmental factors and tick host community may shape microbial interactions (*Elliot, Blanford & Thomas, 2002*). Second, the model assumes that the explanatory variables affect the microbial community composition (or rather, the presence or absence of individual OTUs), but not vice versa. However, this is a valid assumption for most environmental (e.g., elevation and temperature) and tick-related variables (e.g., tick sex, life stage) included in our models. Thirdly, covariation among explanatory variables poses a problem to any correlative modelling approach. Our model is built on two distinct variable sets to aid in handling such covariation: the full variable set includes elevation, whereas the variables with the strongest covariation (i.e., temperature and precipitation) are included in the variable selection set. Fourthly, the inferred residual associations between focal taxa are assumed to be symmetrical. If there are asymmetric interactions (e.g., predator–prey-relationships), the sum outcome can be seen as either positive or negative correlation (*Zurell, Pollock & Thuiller, 2018*). However, in our study, the expectation was facilitation or competition, which are symmetric positive or negative interactions, respectively. Thus, given sufficient signal, we expect that the focal interactions can be captured by our modelling approach.

## CONCLUSIONS

Our study demonstrates that a JSDM framework can contribute to a better understanding of the factors shaping bacterial communities in natural populations as well as patterns of co-occurrence among microbes. Overall, our study highlights the role of small-scale, tick-level characteristics rather than large-scale ecological variation in shaping microbial communities of *I. ricinus*. We identified a number of ecological variables that predict the occurrence of specific tick endosymbionts and human, domestic animal or wildlife pathogens with strong statistical support, but these effects were typically microbe-specific rather than universal. This highlights that environmental change can have different, even opposite effects on different human pathogens, and thus disease risk. Furthermore, by accounting for shared environmental preferences, our approach identified patterns of microbial co-occurrence that are consistent with microbe-microbe interactions, which result in pathogen co-infections within ticks, as well as competition between *Spiroplasma* and a number of human, domestic animal or wildlife pathogens. The latter opens up new and exciting avenues for the control and management of tick-borne diseases in regions with high human disease incidence.

## ACKNOWLEDGEMENTS

We thank Mélissa Lemoine for providing ticks, the numerous people who helped collecting ticks in the field, Glauco Camenisch, Elisa Granato, Jennifer Morger and Alessia Pennachia for help with laboratory work, Lucy Poveda for help with MiSeq sequencing, Frédéric Guillaume for providing access to IT infrastructure, Janine Bolliger and Dirk Schmatz from Swiss Federal Institute for Forest, Snow and Landscape Research WSL for providing

spatial data and for help with spatial analyses, and Otso Ovaskainen for discussions on the modelling and reviewers for constructive comments on the manuscript.

### Funding

This study was funded by Finnish Cultural Foundation Postdoc Pool grant (to Tuomas Aivelo), the Stiftung für wissenschaftliche Forschung an der Universität Zürich (17_027), the Swiss National Science Foundation (PP00P3_128386 and PP00P3_157455), the University of Zurich Research Priority Program "Evolution in Action: from Genomes to Ecosystems", the Faculty of Science of the University of Zurich, and the Baugarten Stiftung (all to Barbara Tschirren). The funders had no role in study design, data collection and analysis, decision to publish, or preparation of the manuscript.

### Grant Disclosures

The following grant information was disclosed by the authors:
Finnish Cultural Foundation Postdoc Pool grant.
Stiftung für wissenschaftliche Forschung an der Universität Zürich.
Swiss National Science Foundation: PP00P3_128386, PP00P3_157455.
University of Zurich Research Priority Program.
Faculty of Science of the University of Zurich.
Baugarten Stiftung.

### Competing Interests

The authors declare there are no competing interests.

### Author Contributions

- Tuomas Aivelo conceived and designed the experiments, performed the experiments, analyzed the data, prepared figures and/or tables, authored or reviewed drafts of the paper, approved the final draft.
- Anna Norberg analyzed the data, prepared figures and/or tables, authored or reviewed drafts of the paper, approved the final draft.
- Barbara Tschirren conceived and designed the experiments, performed the experiments, analyzed the data, contributed reagents/materials/analysis tools, prepared figures and/or tables, authored or reviewed drafts of the paper, approved the final draft.

### Field Study Permissions

The following information was supplied relating to field study approvals (i.e., approving body and any reference numbers):

The field collection was approved by the Amt for Lebensmittelsicherheit und Tiergesundheit des Kantons Graubünden (Dr. T. Bürge and Dr R. Hannimann; permit 2012_17) and the Gemeindeschreiber of the municipalities of Klosters, Grüsch, Rothenbrunnen / Tomils, Flims, Chur / Churwalden and Malans.

## DNA Deposition

The following information was supplied regarding the deposition of DNA sequences:
Sequences are available at the Sequence Read Archive: BioProject PRJNA506875.

## Data Availability

The complete metadata of the samples and their matching sequence accession numbers area available at Figshare:Aivelo, Tuomas; Tschirren, Barbara; Norberg, Anna (2019): Tick data used in ''Human pathogen co-occurrence in Ixodes ricinus ticks: effects of landscape topography, climatic factors and microbiota interactions''. figshare. Dataset. DOI: 10.6084/m9.figshare.7380767.v3.

## Supplemental Information

Supplemental information for this article can be found online at http://dx.doi.org/10.7717/peerj.8217#supplemental-information.

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
