# Peer review of "Bacterial microbiota composition of Ixodes ricinus ticks: the role of environmental variation, tick characteristics and microbial interactions"

_PeerJ, doi:10.7717/peerj.8217_

## Round 0.1 · original submission · Major Revisions

Please provide a detailed point-by-point rebuttal letter along with your revised manuscript. I would appreciate it if you could submit your revision within the next 30 days. In case you need more time please inform me.

Reviewer 1 ·

Basic reporting

Basic Reporting

The manuscript is well written and uses well-referenced publications which make the manuscript coherent, comprehensive and theoretically solid. The manuscript structure accomplishes with Peer J standards norms. I am happy to recommend acceptance however some aspects needs to be considered before its publication. I offer a specific line by line suggestion below:

Supplementary File

1. I found that some paragraphs are repetitive with results (i.e: in page 5 " Using a variance partitioning approach…explaining on average 14.3% of the total variation (Figure 2 in the main text)"; and in page 6 and 7: "The effect sizes varied…to 27%44% at north-facing sites”). I suggest the authors to compile the information in just one section preferably in results.

2. Figures and Tables are relevant and high quality, however, I have found some details that need further revision before publication:

- Figure S3, needs to be re-label. I have found in the supplementary file references to Figure S3a-i, and Figure S3e, but these labels are not included in the Figure S3. I also recommend to include this labeling information in Figure S3 Footnote.

- The authors should check the labeling of Table S2 in the main manuscript. As you can see in Line 289-290 it is referenced the Table S5 instead of Table S2.

3. The reference Lemoine et al. (2018) is included in the narrative but not in the reference list. However, it is included in the reference list of the main manuscript.

Main Manuscript

1. The authors included several times across the manuscript the importance of ticks vectoring pathogens to humans and wildlife. However, I suggest to include the term "domestic animals" as well.

2. In line 177: I recommend to use “reads” instead “amplicons”, and so on across the manuscript.

3. In Line 203, add the web associated with the oligotyping pipeline software (http://oligotyping.org).

4. The paragraph from Line 221 to Line 231 it would probably fit better as part of a conclusion or a discussion. Thus I recommend the authors to consider deleting it from Mat and Met section.

5. In Line 249, please explain which criteria were used to exclude OTUs from the analysis.

6. To facilitate readers the understanding of the analysis and the results I recommend the authors to keep uniformity between the names of the variables in the main manuscript and the Table S1. For example:

- Line 275, instead of Topography use Aspect
- Line278, instead of Tick density use tick abundance since abundance estimates and density estimates are different.

7. In Line 412 and 413, delete the extra parenthesis ).

8. In Line 314 correct Ross et al. (2017), the correct reference is Ross et al. (2018), and in the Reference list as well.

10. I have found an incongruence in the manuscript that needs to be clarified in the Discussion section. In Line 311 to 314 the authors discussed that " This substantial microbiota variation across individual ticks…may be partly explained by some of the bacteria present in ticks being no resident (i.e., bacteria that accidentally obtained from the environment)". However according to the pipeline explained in the Mat and Met section, the authors have previously used methods to get rid of rare OTU (Line 249 "after excluding rare OTUs) and potential environmental contamination and potential environmental contamination (Line 143-144: "To avoid environmental contamination…".). Following this assumption, the method utilized to avoid rare OTUs and potential environmental contamination does not work?.

In this regard, I recommend to include biological explanations about these results and biological expectations can be made from "non-resident bacteria" driving the microbiota variation across individual ticks? I suggest the authors some literature that can be evidenced the importance of the environment in the internal shape and assemblage of tick microbial communities due to bacteria can be eventually acquired from soil and plants from the environment through the spiracles, the mouth and the anal pore, or even during copulation (paternal transmission route) and on each blood-feeding event (skin microbes, pathogens).
Horner-Devine, M.C. and Bohannan, B.J.M. (2006) Phylogenetic clustering and over dispersion in bacterial communities. Ecology 87, S100-S108
Engel, P. and Moran, N.A. (2013) The gut microbiota of insects-diversity in structure and function. FEMS Microbiol. Rev. 37, 699-735
Zolnik, C.P. et al. (2016) Microbiome changes through ontogeny of a tick pathogen vector. Mol. Ecol. 25, 4963-4977
Zolnik, C.P. et al. (2018) Transient influence of blood meal and natural environment on blacklegged tick bacterial communities. Ticks Tick Borne Dis. 9, 563-572
11. In line 42 it should be added “Similarly, the host´s immune system can influence colonization success of microbes… I suggest the author to consider microbes-derived factors in the colonization success of bacteria. It is demonstrated that microbes have mechanisms that participate in the host selection process, (e.g: microbial associated signaling patterns (MAMPs)). (see references)
Chu, H. and Mazmanian, S. K. (2013) Innate immune recognition of the microbiota promote host-microbial symbiosis. Nat. Immunol. 14, 668-675
Lee, S. M. et al. (2013) Bacterial colonization factors control specificity and stability of the gut microbiota. Nature 501, 426-529
12. I suggest the author be more specific about the co-occurring concept (Lines 427-430) What are co-occurring individuals? Where are they co-occurring and why does this matter? it remains complexity. Microbes occurrence with other microbes is based on the assumption that coexisting species are ecologically similar (Leibold and McPeek, 2006, Barberán et al., 2012). But it is relevant to note that co-infections may results when hosts are independently infected by different microorganisms at the same time, or when interaction among parasite species facilitate co-occurrence.

Experimental design

Ticks are well-known as vectors of a great variety of infectious agents but recent surveys have also underlined that the structure of ticks microbial communities shows obvious co-occurrence patterns of non-infectious microbes and pathogens. This interesting because some of these microbes are even essential for tick survival and pathogen competence. Aivelo et al. interestingly discuss the intervention of ecological factors, host characteristics, and interactions among microbes shaping ticks microbial communities. The authors are providing a conceptual framework, which I found sufficiently interesting for publication. I especially liked that the authors included the supplementary file and the raw data with information to be reproducible by other authors. However, I offer specific line by line suggestion below that could help to improve the manuscript:

1. I thank the authors for providing the metadata ("ticksampledata_figshare.csv") available in FigShare, needs to include detailed information of metadata identifiers.
In this regard, I recommend the authors to include details about meaning, and metrics on each case to easily replicate the methods. For example:
- site_masl= Elevation of gradient (meters above sea level).
- Tick_het= tick individual heterozygosis

5. In the Supplementary File, I suggest the authors include in Table S1 more information to be useful to future readers, for example in can be improved as follows:

- Indicate which components and variables are considered "fixed effects" in the same way that Latent variables are considered "random effects".

- Within the “Full variable set”, indicate which variables are considered at “tick-level” or “site-level”.

Validity of the findings

1.While the paper needs improvements, as various definitions and concepts are diffuse and need to be fully developed to finally validate the findings (as I have mentioned above in the Basic Reports).

2. If there is a weakness, it is the small number of ticks samples in some sites, thus a low number of samples and replication, Considering inter- and intraindividual differences and high variation of microbiota the statistical power of analysis for any comparative analyses deserves some important discussions.

3. I suggest the authors revise the "ticksampledata_figshare.csv" file and the ID codes concordance with the column "tick-stageF" , and Table 1, and the total number of Females and Males. If I assume that 0 correspond to males, and 1 to females (similarly to the column tick-stageN in which 0 is adult and 1 nymph). In the column (tick-stageF”) is assigned 0 to nymphs (HXX) and thus considering nymphs (0) as males (0) when filtering the column by 0. This was very confusing to me, and since this was the input matrix used for the HMSC, I am concerned this fact could have influenced the results and conclusions

Reviewer 2 ·

Basic reporting

The manuscript entitled “Bacterial microbiota composition of Ixodes ricinus ticks: the role of environmental variation, tick characteristics and microbial interactions” may be of interest to the readership of PeerJ journal. The research quality and statement conform to the standards and scope of the journal. The study is focused on the identification of the factors driving the taxonomic composition of the microbiota of Ixodes ricinus, including the presence of tick-borne pathogens, which constitutes a frontier of knowledge, with relevance for the prevention of tick-borne diseases. The text is written clearly, with adequate structure. In the introduction, statement and references are correctly used to describe the scientific problem and its background. The methods are clearly described. The data is available online, with the appropriate description. However, I consider that some aspects should be revised:

Mayor comment

1-The introduction should be shortened for greater conciseness.

2- Figure 1 (sampling sites) should be improved. The elevational gradient is the main reason to choose the sampling areas, however, it is not possible to identify it without observing Table 1. To solve this problem, the authors could use different shapes or colors to differentiate low, medium and high elevations points within each location). Remove the unidentified black point at location 1.

3- The major aim of this study is to evaluate the taxonomic composition of microbiota in individual ticks in different conditions, and this is the variable studied in the model. However, only the most abundant OTUs and their distribution are described and included in the results. Better characterization and visualization of all the bacterial diversity should be achieved. For example, a barplot with the taxonomic composition of the samples, grouped by sex and life stage across the elevation gradient should be included. Other compositional traits analysis of the microbiota could be added to complement the analysis, such as alpha and beta diversities.

Experimental design

The study sought to determine how do large-scale abiotic factors and small scale tick-level variables affect tick microbiota composition, and which large-scale abiotic and small-scale tick-level variables predict pathogen occurrence. The authors conducted a well-designed observational study, consistent with the research questions. They chose an elevation gradient that allowed them to work on a well-characterized environmental gradient. They selected standardized framework (JSDM model) to determine the effect of the predictive variables on the taxonomic composition of the microbiota tick. The identification and use of the predictors were well-executed, and individual ticks were used as a random effect in the model which is a strength of this study. As response variable they used the taxonomic composition of the microbiota, assessed through high-throughput amplicon sequencing of the 16S rRNA gene. They conducted a considerable sampling effort, repeated three times (monthly) in each area. However, some aspect of the analytical procedures should be reviewed and improved as possible, specifically related to the variable response on analysis.

Mayor comment

1-The taxonomic composition matrix of all the ticks was used as the response variable (95 total) in the study. However, analyzing the distribution of samples across life stages, sex, sampling area, and elevation, in many of these conditions there were no biological replicates (Table 1), or only two replicates, which could be collected at different times (June, July, August). This suggests that this dataset lacks robustness to support the predictive model used, how did the authors address this limitation? This is an important point because it affects the strength of the conclusions.

2- It was intended to evaluate the effect of the predictors on the taxonomic composition. However, as the authors referred, it is well-known that the microbiota of Ixodes ricinus, as other ticks, is highly variable in taxonomic composition. Even the results of this analysis indicated that more than 60% of the taxonomic composition was explained by the random effect of each individual tick. Perhaps the authors should analyze, at a previous stage, other features of the taxonomic composition of the microbiota less sensitive to changing taxa (i.e. the alpha and beta microbial diversity metrics), this would improve the results.

Validity of the findings

The results are consistently presented and discussed in context.

Mayor comment

1-In the conclusions it is argued that the JSDM framework can contribute to a better understanding of the factors shaping bacterial communities. However, there was no analysis for the evaluation and/or validation of this model. Although its main limitations (in materials and methods) are described, its performance in the analyzed data is not addressed and discussed.

Additional comments

In consequence of the sampling effort made and the importance given to the taxonomic composition in this study, a higher resolution taxonomic classification should be achieved. That is using Qiime2 pipeline since it works at the level of a single nucleotide and each amplicon variant (ASVs) can be identified, improving the taxonomic classification.

·

Basic reporting

The manuscript by Aivelo et al. investigates the effect tick-specific and environmental variables have on the microbiome composition of Ixodes ricinus. In particular the manuscript looks at the impact of these variables on selected endosymbiont and pathogenic bacteria. The authors are to be commended on the inclusions of extensive supplementary methodology and details on modelling. While it is clear the authors had a large amount of data to work with they have done well to be selective in their inclusive of results to include data which makes the most sense from an ecological point of view. The manuscript is well written with clear aims.

The authors have made the raw data available on the NCBI Sequence Read Archive. However 93 samples are listed under the designated BioProject PRJNA506875. The manuscript states that 92 tick samples were sequence (no controls sequenced), please include the BioSamples that were used in the present study (e.g. BioSample SAMNXXXXXXXX – SAMNXXXXXXXX). The authors have made additional data available on FigShare repository.

Experimental design

The authors make use of ecological modelling methods in the context of tick-associated bacteria, including pathogens. The authors are to be commended on the detailed explanation that is made available in the supplementary information. In particular references regarding their classification of bacteria as endosymbiont or pathogen. This is a challenging and contentious aspect of tick microbiology that many authors do not justify. The authors have also given extensive detail on modelling aspects used in the manuscript.

Validity of the findings

Absence of sequences from negative controls in 16S NGS studies is concerning and raises questions as to the use of appropriate controls. Evidence of both (i) bacteria present in extraction kits known as the and (ii) unavoidable cross-contamination during sample processing and DNA extraction of ticks, are well reported tick microbiome and general microbiome/eDNA studies. (E.g. Couper et al. 2019, DOI: 10.1002/ece3.5361; Gofton et al. 2015, DOI: 10.1186/s13071-015-0958-3). Can the authors please confirm if controls were included in sequencing libraries?

Taxonomy of OTUs – while the authors have provided detailed additional information in the supplementary information there is little details on the taxonomic assignment of OTUs other than the following sentence “We used 97% similarity to determine OTUs and classified them with the wang method (Wang et al. 2007) using SILVA taxonomy.”. As the authors have assigned species level taxonomy to OTUs (table 2) can they please expand on how they assigned this level of taxonomy (e.g. BLAST confirmation and results?). In the case of Rickettsia spp. in particular the accurate taxonomic assignment through a short 250 bp amplicon of the 16S rRNA gene is extremely difficult (Greay et al. 2018 10.1186/s13071-017-2550-5; Raoult and Roux 1997 Clin Microbiol Rev 10(4). If the authors used additional taxonomic assignment analysis or confirmed by other molecular tools can they please expand.

Additional comments

[1] Line 54-55 – The number of OTUs is highly ambiguous, and influenced by numerous factors including target amplicon, bioinformatics and library preparation (Youssef et al. 2009, DOI: 10.1128/AEM.00592-09; Chen et al 2013, DOI: 10.1371/journal.pone.0070837). As such reference reporting on a designated number of OTUs in a specific tick species is misleading. Unless the authors wish to expand on comments about the number of OTUs in ticks including biases involved, this should be removed.

[2] Line 160– The use of the word “quantified” is inaccurate as the authors did not quantify bacterial load (via qPCR or otherwise). The number of 16S NGS reads is not directly relative to bacterial quantification in a community, unless attempts to account for such biases were made (e.g. Kembel et al. 2012, DOI: journal.pcbi.1002743, Rosselli et al. 2015, DOI: 10.1038/srep32165). Unless these methods were implemented it is recommended wording be changed to “characterisation of bacterial community” or “profiling of the tick bacterial community” or similar.

[3] Line 160 – Reference to the V3-4 region of the 16S gene is incorrect if primers 515/806 were used as is stated in the manuscript. This primer pair amplifies only the V4 region (see http://www.earthmicrobiome.org/protocols-and-standards/16s/).

[4] Lines 177 – throughout manuscript - The authors use the words amplicons and sequences interchangeably. If they are referring to different aspects can they please clarify this clearly, otherwise the authors should ensure wording is consistent throughout the manuscript.

[5] Line 180-181 – A threshold of >5 amplicons is extremely low, can the authors please justify this? Without any sequence data from controls, the level of contamination or cross-talk is difficult to assess.

[6] Line 236-237 – See above for comments regarding no amplification from controls.
References – some references are lacking italics for species names e.g. lines 472-474, 480.

---

## Round 0.2 · accepted · Accept

Thank you for the thorough revision.

Reviewer 1 ·

Basic reporting

no comment

Experimental design

no comment

Validity of the findings

no comment

Additional comments

Considering that the authors have complied with the feedback from the reviewers and they have modified it accordingly, I am happy to recommend acceptance of the manuscript.

Reviewer 2 ·

Basic reporting

The manuscript was improved.

The authors considered the suggestions and responded convincingly to my comments, and in my opinion, to the comments of the other reviewers as well.

Experimental design

The description of the materials and methods was improved.

Validity of the findings

The authors achieved a correct presentation of their results, with adequate contextualization of the strengths and weaknesses of the study, and their contribution to the research area.

·

Basic reporting

The manuscript is well written and would be of interest to Peerj readers and those in the field of tick research. Sequence data is provided on the NCBI Sequence Read Archive and supplementary information made available.

Experimental design

The authors have provided extensive details on methodology, made available in the supplementary information. I am happy that they have addressed reviewer comments regarding experimental design.

Validity of the findings

The results are presented in a consistent manner. Any limitations in the data have been discussed and they have incorporated reviewer comments.

Additional comments

The authors have thoroughly addressed all reviewer comments, and I am pleased to recommend the manuscript be accepted.